# Optimal Error Quantification and Robust Tracking under Unknown Upper Bounds on Uncertainties and Biased External Disturbance

**Victor F. Sokolov**

Institute of Physics and Mathematics, Federal Research Center Komi Science Center, Ural Branch, RAS, 167982 Syktyvkar, Russia; sokolov@ipm.komisc.ru

**Abstract:** This paper addresses a problem of optimal error quantification in the framework of robust control theory in the $\ell_1$ setup. The upper bounds of biased external disturbance and the gains of coprime factor perturbations in a discrete-time linear time invariant SISO plant are assumed to be unknown. The computation of optimal data-consistent upper bounds under a known bias of external disturbance has been simplified to linear programming. This allows for the computation of optimal estimates in real-time and their application to achieve optimal robust steady-state tracking even when facing an unknown bias in the external disturbance. The presented results have been illustrated through computer simulations.

**Keywords:** robust tracking; error quantification; model evaluation; optimal control; adaptive control

**MSC:** 93D21; 49K35

## 1. Introduction

The present paper addresses a specific problem of optimal error quantification for robust control. Robust control deals with systems under uncertainties and external disturbances. The theory of robust control emerged in the 1980s, and by the mid-1990s, basic results on robust stability and robust performance had been obtained [1]. Robust control synthesis requires not only a nominal model, but also a quantification of the uncertainty in that model [2]. Problems of finding a nominal model and error quantification were recognized as the main problems of system identification for practical applications of the robust control theory [3,4]. These problems have remained an open issue until the present time both in online and offline settings [5,6]. The main distinction arises from deterministic models of external disturbances and uncertainties in robust control theory, as opposed to the predominant use of stochastic external disturbances and the absence of uncertainties in identification theory.

The model of deterministic external disturbances is used in system identification under the set-membership approach, where upper bounds on magnitudes of disturbances are assumed to be known a priori [7]. The assumption of known upper bounds was criticized for "conservatism of bounds, sensitivity of bounds to accuracy of prior information, and difficulty in providing prior information" [2]. It was noted in [8] that "The activity on estimation of uncertainty sets was often erroneously put under the umbrella of identification for control, since in most of this work the control objective was not taken into account in the identification design". Difficulties in accounting for control objectives when quantifying errors are inevitable in $H_\infty$ robust control theory, a dominant area of robust control that corresponds to the $\ell_2$ signal space. This challenge arises because direct representations for control criteria have not been derived in $H_\infty$ theory. Consequently, problems related to optimal error quantification in $H_\infty$ theory have either never been considered or have

been addressed using artificial identification criteria, such as minimizing a convex linear combination of upper bounds on uncertainty and disturbance [9,10].

The traditional set-membership approach to system identification under known upper bounds of disturbances excludes problems of error quantification. It was used mainly for computation of various upper and lower polyhedral, ellipsoidal or other estimates of the sets of unknown control system parameters consistent with measurement data. Rare applications of these estimates to control problems are of a heuristic nature and do not have a strict mathematical background. Nonconservative and strictly proven stabilization of the linear time-invariant plant with an unknown matrix of the state equation under bounded external disturbance and measurement noise with unknown upper bounds was recently proposed without the direct use of any identification [11]. In this paper, set estimation and error quantification are "hidden" in the $\ell_1$ optimization problem associated with a linear representation of the current state $x_t$ through the previous states $x_0, \cdots, x_{t-1}$ and require additional storage of the previous controls $u_0, \cdots, u_{t-1}$.

The present paper addresses the problem of optimal error quantification in the framework of the $\ell_1$ theory of robust control, which corresponds to the model of bounded external disturbances and the $\ell_\infty$ signal space. In the $\ell_1$ robust control theory, basic results on robust stability and robust performance were presented in [12,13] and direct representations for the steady-state tracking error were obtained in [14–16]. These results opened the door for using the control criterion as the identification criterion in problems of error quantification and synthesis of adaptive optimal control [17]. The problem of model evaluation on the basis of error quantification for robust steady-stated tracking was considered in [18] in an offline setting for the discrete-time plant, with the linear time invariant nominal model under coprime factor perturbations, bounded external disturbance, and measurement noise with unknown upper bounds.

In the present paper, the problem of error quantification is considered in the more advanced optimal setup. Upper bounds of the gains of coprime factor perturbations and biased external disturbance in a discrete-time minimum phase plant are assumed to be unknown. The control criterion is the worst-case upper bound of the steady-state tracking error. This criterion is proven to be linear-fractional with respect to the unknown parameters. The solution of the problem under a known bias of the external disturbance is based on treating the control criterion as an identification criterion and on recursive computation of polyhedral estimates, consistent with measurement data, of unknown upper bounds and gains. Since linear-fractional programming is reducible to linear programming [19], the optimal error quantification becomes computationally tractable online. Polyhedral estimates of unknown parameters are described by linear inequalities with respect to these parameters. Due to the choice of a sufficiently small dead zone parameter, the optimal error quantification is solved with a prescribed accuracy and the number of possible updates of the polyhedral estimates is finite.

The remaining contents of the paper are organized as follows. Key notation is described at the end of the Introduction. A model description and a preliminary problem statement are presented in Section 1. Representation for the steady-state tracking error of the optimal closed loop system in the framework of the $\ell_1$ theory of robust control is derived in Section 3. The problem of optimal error quantification under a known bias of the external disturbance is strictly formulated in Section 4 and its solution with a prescribed accuracy is described in Section 5. Section 6 presents simulations and comments on them. Robust steady-state tracking under an unknown bias of the external disturbance is discussed in Section 7. Section 8 concludes the paper.

**Notation 1.** $|\varphi|$ − *the Euclidean norm of the vector* $\varphi \in \mathbb{R}^n$;
$\ell_e$ − *the vector space of real sequences* $x = (\cdots, x_{-2}, x_{-1}, x_0, x_1, x_2, \cdots)$,
$x_s^t = (x_s, x_{s+1}, \ldots, x_t)$ *for* $x \in \ell_e$;
$|x_s^t| = \max_{s \le k \le t} |x_k|$;
$\ell_\infty$ − *the normed space of bounded real sequences* $x = (x_0, x_1, x_2, \ldots)$,

$\|x\|_{\infty} = \sup_t |x_t|$ *for* $x \in \ell_{\infty}$;

$\|x\|_{ss} = \limsup_{t \to +\infty} |x_t|$,

$\ell_1$ – *the normed space of absolutely summable sequences,*

$\|x\|_1 = \sum_{k=0}^{+\infty} |x_k|$ *for* $x \in \ell_1$;

$\|G\| = \sum_{k=0}^{+\infty} |g_k| = \|g\|_1$ – *the induced norm of stable linear time-invariant system* $G : \ell_{\infty} \to \ell_{\infty}$ *with the transfer function* $G(\lambda) = \sum_{k=0}^{+\infty} g_k \lambda^k$.

## 2. Model Description and Preliminary Problem Statement

Let the model of the controlled discrete-time plant be described by equation

$$a(q^{-1})y_t = b(q^{-1})u_t + v_t, \quad t = 1, 2, 3, \ldots, \tag{1}$$

where $y_t \in \mathbb{R}$ is the output at the time instant $t$, $u_t \in \mathbb{R}$ is the control, $v_t \in \mathbb{R}$ is the total disturbance,

$$a(q^{-1}) = 1 + a_1 q^{-1} + \ldots + a_n q^{-n}, \ b(q^{-1}) = b_1 q^{-1} + \ldots + b_m q^{-m}, \ b_1 \neq 0,$$

and $q^{-1} : \ell_e \to \ell_e$ is the backward shift operator ($q^{-1}y_t = y_{t-1}$). Initial values $y_{1-n}^0 = (y_{1-n}, \ldots, y_0)$ are arbitrary, $y_k = 0$ for $k < 1 - n$, and $u_k = 0$ for $k < 0$.

The polynomials $a(\lambda) = 1 + a_1 \lambda + \ldots + a_n \lambda^n$ and $b(\lambda) = b_1 \lambda + \ldots + b_m \lambda^m$ characterize the known nominal model of the plant and the roots of $b(\lambda)/\lambda$ are outside the closed unit disk of the complex plane (such a plant is called minimum phase).

The total disturbance $v_t$ in (1) is modeled in the form accepted in the $\ell_1$-theory of robust control

$$v_t = c^w + \delta^w w_t + \delta^y \Delta^1(y)_t + \delta^u \Delta^2(u)_t \ \forall t, \quad \|w\|_{\infty} \leq 1. \tag{2}$$

In (2), $c^w + \delta^w w_t$ describes the bounded external disturbance, where $c^w$ is its bias, $w \in \ell_{\infty}$ is an unknown normalized real sequence, $\delta^w$ is the upper bound of unbiased disturbance. The last two terms in (2) represent coprime factor perturbations satisfying inequalities

$$|\Delta^1(y)_t| \leq |y_{t-\mu}^{t-1}|, \quad |\Delta^2(u)_t| \leq |u_{t-\mu}^{t-1}| \ \forall t. \tag{3}$$

The real numbers $\delta^y$ and $\delta^u$ are the gains (i.e., the $\ell_{\infty}$-induced norms) of uncertainties in output and control, respectively. Normalized uncertainties $\Delta^i : \ell_{\infty} \to \ell_{\infty}, i = 1, 2$ are unknown strictly causal nonlinear time-varying operators [12]. The parameter $\mu$ in (3) describes the memory of the uncertainties $\Delta^1$ and $\Delta^2$ and can be chosen by the controller designer to be sufficiently large without introducing excessive conservatism into the disturbance model (see Comment 1 to Theorem 1 in Section 3). One can show (see Lemma 4 [14] and Lemma 1 [17] for details) that the description of the total disturbance $v$ in the form (2) and (3) is equivalent to the description

$$|v_t - c^w| \leq c^w + \delta^y |y_{t-\mu}^{t-1}| + \delta^u |y_{t-\mu}^{t-1}| \ \forall t. \tag{4}$$

*Preliminary problem statement.* The vector

$$\delta = (\delta^w, \delta^y, \delta^u)^T$$

of the parameters of the total disturbance $v$ is assumed to be *unknown* and the problem under consideration is to provide the online optimal error quantification in the framework of the robust control theory in the $\ell_1$ setup.

## 3. Optimal Robust Tracking under Known Nominal Model

Let us introduce the notation $\xi = (a_1, \ldots, a_n, b_1, \ldots, b_m)^T$ for the vector of known parameters of the nominal model.

Let $r \in \ell_\infty$ be a given bounded reference signal and the control criterion is the worst-case steady state tracking error

$$J_\mu(\delta) = \sup_{v \in V} \limsup_{t \to +\infty} |y_t - r_t| = \sup_{v \in V} \|y - r\|_{ss}, \tag{5}$$

where sup is computed on the set $V$ of the total disturbances $v$ of the form (2), (3). Note that specific values $v_t$ of the total disturbance $v$ are determined by specific admissible values $w_t$, $\Delta^1(y)_t$, and $\Delta^2(u)_t$.

Consider the controller described by the equation

$$b(q^{-1})u_t = (a(q^{-1}) - 1)y_t + r_t - c^w. \tag{6}$$

Then we have for the tracking error in the closed loop system (1) and (6)

$$y_t - r_t = v_t - c^w = \delta^w w_t + \delta^y \Delta^1(y)_t + \delta^u \Delta^2(u)_t. \tag{7}$$

Due to the unpredictability and arbitrariness of the values of the right-hand side in (7) at the time of computing the control $u_t$, it follows that the controller (6) is optimal for the control criterion (5).

Let us introduce notation for the transfer function from $y$ to $u$ of the controller (6)

$$G_{uy}^{\xi}(\lambda) = \frac{a(\lambda) - 1}{b(\lambda)}$$

and rewrite the controller equation (6) as follows

$$u_t = G_{uy}^{\xi}(q^{-1})y_t + \frac{1}{b(q^{-1})}r_t - \frac{1}{b(q^{-1})}c^w. \tag{8}$$

**Definition 1.** *The closed loop system (1) and (6) is said to be robustly stable if $J_\mu(\delta) < +\infty$.*

**Definition 2.** *We will say that the sequence $|r|$ gets into neighborhoods of $\|r\|_{ss}$ uniformly often, if for arbitrary $\varepsilon > 0$ there exists $T > 0$ and a sequence $t_1 < t_2 < t_3 < \dots$ so that*

$$\forall j \in \mathbb{N} \quad 0 < t_{j+1} - t_j \leq T \wedge |r_{t_{j+1}}| \geq \|r\|_{ss} - \varepsilon.$$

*The performance of the optimal closed loop system (1) and (6) is described in the next theorem.*

**Theorem 1.** *The following statements are true for the closed system (1) and (6).*

*(1) The system with infinite memory (i.e., $\mu = +\infty$) perturbations is robustly stable if and only if*

$$\delta^y + \delta^u \|G_{uy}^{\xi}\| < 1. \tag{9}$$

*For the system with $\mu = +\infty$ and zero initial data $y_{1-n}^0$*

$$J(\delta) := J_{+\infty}(\delta) = \frac{\delta^w + \delta^y \|r\|_{ss} + \delta^u (|c^w| + \|r\|_{ss}) \|1/b(q^{-1})\|}{1 - \delta^y - \delta^u \|G_{uy}^{\xi}\|}. \tag{10}$$

*(2) For the system with $\mu < +\infty$ and arbitrary initial data $y_{1-n}^0$,*

$$J_\mu(\delta) \leq J_{+\infty}(\delta) \quad \forall \mu > 0. \tag{11}$$

*If the sequence $|r|$ gets into neighborhoods of the limsup $\|r\|_{ss}$ uniformly often, then for any initial data*

$$J_\mu(\delta) \nearrow J_{+\infty}(\delta) \quad (\mu \to +\infty), \tag{12}$$

*where the sign $\nearrow$ means the monotonic convergence from below as $\mu \to +\infty$.*

**Proof.** To prove Theorem 1, we represent the closed loop system (1), (6) in the standard $M - \Delta$ form shown in Figure 1 and described by equations

$$\begin{pmatrix} e \\ z \end{pmatrix} = M \begin{pmatrix} f \\ w \\ \xi \end{pmatrix}, \quad e = y - r, \quad \xi = \Delta z.$$ (13)

where $z$ and $\xi$ are, respectively, the input and output of the structured uncertainty $\Delta$,

$$z_t = \begin{pmatrix} y_t \\ u_t \end{pmatrix}, \xi = \begin{pmatrix} \Delta^1 & 0 \\ 0 & \Delta^2 \end{pmatrix} z = \begin{pmatrix} \Delta^1(y) \\ \Delta^2(u) \end{pmatrix},$$

and $f$ is a fixed input signal including the tracked signal $r$ and a constant signal equal to 1 to account for the bias $c^w$:

$$f = \begin{pmatrix} r \\ \mathbf{1} \end{pmatrix}, \quad \mathbf{1} := (1, 1, \dots) \in \ell_\infty.$$

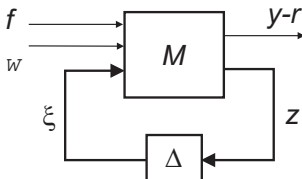

**Figure 1.** M-$\Delta$ form of the system (1), (6).

Let us represent the matrix $M$ in (13) in block form corresponding to the input and output signals in Figure 1:

$$M = \begin{pmatrix} M_{ef} & M_{ew} & M_{e\xi} \\ M_{zr} & M_{zw} & M_{z\xi} \end{pmatrix}.$$ (14)

For the system (1), (6) this presentation is of the form

$$M = \begin{pmatrix} 0 & 0 & \delta^w & \delta^y & \delta^u \\ 1 & 0 & \delta^w & \delta^y & \delta^u \\ \dfrac{1}{b(q^{-1})} & -\dfrac{c^w}{b(q^{-1})} & \delta^w G_{uy}^\xi & \delta^y G_{uy}^\xi & \delta^u G_{uy}^\xi \end{pmatrix},$$ (15)

where $q$ is the forward shift operator ($qr_t = r_{t+1}$). The first row of the matrix $M$ in (15) corresponds to the right-hand side of the equality (7). The second row of $M$ is obtained by moving $r_t$ to the right-hand side of the equality (7). The third row of $M$ corresponds to the representation of the optimal controller in the form (8).

The necessary and sufficient condition for robust stability (9) follows from Theorem 7 in [16] applied to the system (1), (6).

To prove the representation (10) for the control criterion $J_{+\infty}(\delta)$, it suffices to apply Theorems 5 and 6 of [16]. Let us introduce the notation

$$[A]_1 := \begin{pmatrix} \|A_{11}\|_1 & \cdots & \|A_{1q}\|_1 \\ \vdots & \vdots & \vdots \\ \|A_{p1}\|_1 & \cdots & \|A_{pq}\|_1 \end{pmatrix}$$

for an arbitrary $p \times q$ matrix $A$ of impulse responses $A_{ij} \in \ell_1$.

For the block matrix $M$ in (14), we define

$$M_{ss}(f) := \begin{pmatrix} [M_{ef}f]_{ss} + [M_{ew}]_1 & [M_{e\xi}]_1 \\ [M_{zf}r]_{ss} + [M_{zw}]_1 & [M_{z\xi}]_1 \end{pmatrix}.$$

The matrix $M_{ss}(f)$ for the specific matrix $M$ in (15) takes the form

$$
\begin{pmatrix}
 & \delta^w & \delta^y & \delta^u \\
 & \|r\|_{ss} + |\delta^w| & \delta^y & \delta^u \\
 (\|r\|_{ss} + |c^w|)\|1/b(q^{-1})\| + \delta^w\|G_{uy}^{\zeta}\| & \delta^y\|G_{uy}^{\zeta}\| & \delta^u\|G_{uy}^{\zeta}\|
\end{pmatrix}. \tag{16}
$$

According to Theorem 5 in [16],

$$
\begin{aligned}
J_{+\infty}(\delta) =& [M_{er}r]_{ss} + [M_{ew}]_1 + \\
& [M_{y\zeta}]_1(I - [M_{z\zeta}]_1)^{-1}([M_{zr}r]_{ss} + [M_{zw}]_1).
\end{aligned} \tag{17}
$$

Applying the formula (17) to the matrix (16), we get

$$
J(\delta) = \delta^w + (\delta^y \ \delta^u)\left(I - \begin{pmatrix} \delta^y & \delta^u \\ \delta^y\|G_{uy}^{\zeta}\| & \delta^u\|G_{uy}^{\zeta}\| \end{pmatrix}\right)^{-1} \times
$$

$$
\begin{pmatrix} \|r_{ss}\| + \delta^w \\ (\|r\|_{ss} + |c^w|)\|1/b(q^{-1})\| + \delta^w\|G_{uy}^{\zeta}\| \end{pmatrix} =
$$

$$
\delta^w + \frac{1}{1 - \delta^y - \delta^u\|G_{uy}^{\zeta}\|}(\delta^y \ \delta^u)\begin{pmatrix} 1 - \delta^u\|G_{uy}^{\zeta}\| & \delta^u \\ \delta^y\|G_{uy}^{\zeta}\| & 1 - \delta^y \end{pmatrix} \times
$$

$$
\begin{pmatrix} \|r_{ss}\| + \delta^w \\ (\|r\|_{ss} + |c^w|)\|1/b(q^{-1})\| + \delta^w\|G_{uy}^{\zeta}\| \end{pmatrix}.
$$

Considering that

$$
(\delta^y \ \delta^u)\begin{pmatrix} 1 - \delta^u\|G_{uy}^{\zeta}\| & \delta^u \\ \delta^y\|G_{uy}^{\zeta}\| & 1 - \delta^y \end{pmatrix} = (\delta^y \ \delta^u),
$$

we obtain the representation (10)

$$
\begin{aligned}
J(\delta) =& \delta^w + \frac{1}{1 - \delta^y - \delta^u\|G_{uy}^{\zeta}\|} \times \\
& (\delta^y \ \delta^u)\begin{pmatrix} \|r_{ss}\| + \delta^w \\ (\|r\|_{ss} + |c^w|)\|1/b(q^{-1})\| + \delta^w\|G_{uy}^{\zeta}\| \end{pmatrix} = \\
& \frac{\delta^w + \delta^y\|r\|_{ss} + \delta^u\|r\|_{ss}\|1/b(q^{-1})\|\| + \delta^u|c^w|\|1/b(q^{-1})\|)}{1 - \delta^y - \delta^u\|G_{uy}^{\zeta}\|}.
\end{aligned}
$$

The inequality (11) follows from the fact that the set of perturbations (3) with bounded memory $\mu$ is a subset of perturbations with infinite memory. The monotonicity of $J_\mu(\delta)$ with respect to $\mu$ follows from the enlargement of the sets of perturbations (3) with the increase of $\mu$. Finally, the convergence of $J_\mu(\delta)$ to $J_{+\infty}(\delta)$ is guaranteed by Theorem 6 in [16]. Theorem 1 is proved. □

Theorem 1 provides the guaranteed upper bound $J(\delta)$ for the steady-state tracking error, and this upper bound is tight, e.g., for common periodic or constant reference signals.

**Remark 1.** *The bounded memory perturbation model (3) was proposed in [16] instead of the finite or fading memory perturbation models introduced in [14] because the latter are not testable against data. The model of bounded memory perturbations needs the computation of $|y_{t-\mu}^{t-1}|$ and $|u_{t-\mu}^{t-1}|$ for the purpose of online estimation and this computation, even for large $\mu$, is not a considerable problem for modern computers. The larger the memory size $\mu$ chosen by the control designer, the less conservative the upper bound (11). At the same time, choosing too large values of $\mu$ makes little sense in real applications.*

## 4. Problem of Optimal Error Quantification

The robust stability condition (2) requires the following a priori assumption

**Assumption 1.** *The vector*

$$\delta = (\delta^w, \delta^y, \delta^u)^T$$

*of parameters of the total disturbance $v$ is unknown and satisfies the inequality*

$$\delta^y + \delta^u \|G_{uy}^{\xi}\| \leq \bar{\delta} < 1 \tag{18}$$

*with a known $\bar{\delta}$.*

The number $\bar{\delta} > 0$ is chosen by the controller designer on the basis of a priori information or even without it and can be as close to 1 as desired. Assumption 1 excludes from consideration non-stabilizable models and models that are unacceptable for practical application since they are too close to the boundary of the region of robustly stabilizable models. This is because the value of the control criterion $J(\delta)$ for these models is too large in view of the small denominator in the representation (10).

**Problem 1.** *Let the plant (1) with the known nominal model, bias $c^w$ of the external disturbance, and unknown parameters $\delta$ of the total disturbance $v$ be controlled by the the controller (6). The problem under consideration is to online compute the minimal upper bound of the steady-state tracking error $|y - r|_{ss}$ consistent with the current measurement data $(y_0^t, u_0^{t-1})$ and with the prescribed accuracy.*

## 5. Optimal Online Error Quantification

Let us introduce the notation

$$p_t^y = |y_{t-\mu}^{t-1}|, \quad p_t^u = |u_{t-\mu}^{t-1}|. \tag{19}$$

**Lemma 1.** *Let $\hat{\delta} = (\hat{\delta}^w, \hat{\delta}^y, \hat{\delta}^u)^T$ be some estimate of $\delta$. If*

$$|a(q^{-1})y_t - b(q^{-1})u_t - c^w| \leq \hat{\delta}^w + \hat{\delta}^y p_t^y + \hat{\delta}^u p_t^u, \tag{20}$$

*for all sufficiently large $t$, then the output of the closed loop system (1) and (6) satisfies the inequality*

$$\|y - r\|_{ss} \leq J(\hat{\delta}). \tag{21}$$

**Proof.** Let us define a sequence of imaginary total disturbance

$$\hat{v}_t = a(q^{-1})y_t - b(q^{-1})u_t \quad \forall t.$$

Then the output $y$ satisfies the equation (1) with the total disturbance $\hat{v}_t$ and

$$|\hat{v}_t - c^w| \leq \hat{\delta}^w + \hat{\delta}^y p_t^y + \hat{\delta}^u p_t^u \tag{22}$$

for all sufficiently large $t$ in view of (20). Since violations of the inequalities (22) are possible on finite initial time intervals only, one can apply the second statement of Theorem 1 to the closed loop system with the total disturbance $\hat{v}$ and get the inequality (21). Lemma 1 is proven. □

At time $t$, complete information about the unknown vector $\delta$ in the closed loop system (1) and (6) is in the form of the set estimate

$$\delta \in D_t = \{\hat{\delta} \mid |a(q^{-1})y_k - b(q^{-1})u_k - c^w| \leq \hat{\delta}^w + \hat{\delta}^y p_k^y + \hat{\delta}^u p_k^u \ \forall k \leq t \}.$$

Then the best estimate of $\delta$, unfalsified by data $y_0^t, u_0^{t-1}$, is as follows

$$\delta_t^{opt} = \operatorname*{argmin}_{\hat{\delta} \in D_t} J(\hat{\delta}) . \tag{23}$$

The inequalities in the description of the set estimates $D_t$ can contain redundant inequalities, which can be excluded online (see, e.g., [20]). Despite this, the number of inequalities in the description of $D_t$ can increase without limit as $t \to +\infty$. To avoid an unlimited increase in the number of stored inequalities, we will solve the problem (23) with the prescribed accuracy.

We choose a positive number $\varepsilon$, which should be sufficiently small and characterizes the size of the dead zone when updating the outer set estimates $S_t$ of the unknown vector $\delta$. At each time instant, a set estimate $S_t$ and a vector estimate $\delta_t$ will be computed. The estimates $S_0$ and $\delta_0$ are defined as follows.

$$S(0) := \{ \hat{\delta} = (\hat{\delta}^w, \hat{\delta}^y, \hat{\delta}^u)^T \mid \hat{\delta} \geq 0, \ \hat{\delta}^y + \hat{\delta}^u \|G_{uy}^\xi\| \leq \bar{\delta} < 1 \}, \delta_0 = (0,0,0)^T.$$

We introduce the notation

$$v_{t+1} = |a(q^{-1})y_{t+1} - b(q^{-1})u_{t+1} - c^w|, \quad \phi_{t+1} = (1, p_{t+1}^y, p_{t+1}^u)^T$$

and rewrite the inequality (20) that presents new information about the unknown vector $\delta$ at the time instant $t + 1$ in the form of the inclusion

$$\delta \in \Omega_{t+1} = \{ \hat{\delta} \mid v_{t+1} \leq \hat{\delta}\phi_{t+1} \} . \tag{24}$$

Let $S_t$ and $\delta_t$ be the set and vector estimates of $\delta$ at the time instant $t$, respectively. Define

$$S_{t+1} = \begin{cases} S_t , & \text{if } v_{t+1} \leq \delta_t\phi_{t+1} + \varepsilon|\phi_{t+1}|, \\ S_t \cap \Omega_{t+1} , & \text{otherwise} , \end{cases} \tag{25}$$

$$\delta_{t+1} = \operatorname*{argmin}_{\hat{\delta} \in S_{t+1}} J(\hat{\delta}) . \tag{26}$$

Geometric interpretation of the updating (25) is simple. The set estimate $S_t$ is updated by adding the new inequality from (24) if and only if the distance from $\delta_t$ to the half-space $\Omega_{t+1}$ is greater $\varepsilon$. Note that the vector estimate remains the same, i.e., $\delta_{t+1} = \delta_t$, in the case $S_{t+1} = S_t$.

**Theorem 2.** *Let Assumption 1 be satisfied and the chosen dead zone parameter $\varepsilon$ in (25) satisfy the inequalities*

$$0 < \varepsilon < \frac{1 - \bar{\delta}}{1 + \|G_{uy}^\xi\|} . \tag{27}$$

*Let the set estimates $S_t$ and the vector estimates $\delta_t$ be computed in the closed loop system (1) and (6) according to (25) and (26). Then the number of updates of the set estimates $S_t$ and the vector estimates $\delta_t$ is finite and the output $y$ of the closed loop system satisfies*

$$\|y - r\|_{ss} \leq J(\delta_\infty^\varepsilon) = J(\delta_\infty) + O(\varepsilon) \quad (\varepsilon \to 0) , \tag{28}$$

*where $\delta_\infty = \lim_{t \to +\infty} \delta_t$ is the final value of $\delta_t$,*

$$\delta_\infty^\varepsilon = (1 + \varepsilon, \delta_\infty^y + \varepsilon, \delta_\infty^u + \varepsilon) ,$$

*and*

$$J(\delta_\infty) \leq J(\delta) . \tag{29}$$

**Proof.** Under each updating $\delta_t$, we have from (25)

$$\delta_t \phi_{t+1} < \nu_{t+1} - \varepsilon|\phi_{t+1}|.$$

Then for any $\hat{\delta} \in \Omega_{t+1}$,

$$|\hat{\delta} - \delta_t||\phi_{t+1}| \geq |(\hat{\delta} - \delta_t)\phi_{t+1}| > \varepsilon|\phi_{t+1}|$$

and, therefore, $|\hat{\delta} - \delta_t| > \varepsilon$. This means that the distance from the vector estimate $\delta_t$ to the half-space $\Omega_{t+1}$ is greater than $\varepsilon$. Since $\delta_{t+1} \in \Omega_{t+1}$, the distance between $\delta_t$ and $\delta_{t+1}$ is greater than $\varepsilon$. Then the distance from $\delta_t$ to all future estimates $\delta_k, k > t$ is greater than $\varepsilon$ because $S_k \subset S_t \subset \Omega_{t+1}$. Then the number of updates of $S_t$ and $\delta_t$ cannot be infinite if the sequence of estimates $\delta_t$ is bounded, since each update of set estimates excludes disjoint balls with a radius of $\varepsilon/2$ and with centers at $\delta_t$. The sequence of estimates $\delta_t$ is bounded because the set $\{\hat{\delta} \mid J(\hat{\delta}) \leq J(\delta)\}$ is bounded and $J(\delta_t) \leq J(\delta)$ for all $t$ in view of (26). The inequality (29) follows from (26) and the inclusions $\delta \in S_t$ for all $t$. The inequality in (28) is guaranteed by Lemma 1 for $\hat{\delta} = \delta_\infty^\varepsilon$. Finally, the equality in (28) follows from the inequality

$$|\phi_t| \leq 1 + p_t^y + p_t^u,$$

the definition of $\delta_\infty^\varepsilon$, and the monotonicity of $J(\delta)$ with respect to the components of $\delta$. Theorem 2 is proven. □

**Remark 2.** *The main merit of the presented algorithm (25) and (26) of optimal online error quantification is it that, consistent with the measurement data $y_0^t, u_0^t$, it provides a nonconservative upper bound $J(\delta_\infty^\varepsilon)$ for the steady-state tracking error*

$$\|y - r\|_{ss} \leq J(\delta_\infty^\varepsilon) = J(\delta) + O(\varepsilon) \quad (\varepsilon \to 0),$$

*which follows from (21) and (29) and where the value of $J(\delta)$ corresponding to the "true" vector $\delta$ is not known.*

**Remark 3.** *To guarantee the desired accuracy of error quantification, one can compute the differences*

$$J(\delta_t^\varepsilon) - J(\delta_t), \quad \text{where} \quad \delta_t^\varepsilon = (1 + \varepsilon, \delta_t^y + \varepsilon, \delta_t^u + \varepsilon). \tag{30}$$

*Taking into account the inequality $J(\delta_t) \leq J(\delta)$ (since $\delta \in S_t$ for all $t$, see the proof of Theorem 2), we get*

$$J(\delta_t^\varepsilon) - J(\delta) \leq J(\delta_t^\varepsilon) - J(\delta_t).$$

*If the current difference is greater than the desired accuracy, one can choose a smaller (half as much, for example) dead zone parameter $\varepsilon$. The desired accuracy will be guaranteed after several corrections of $\varepsilon$, if they ever occur.*

**Remark 4.** *Computation of the estimate $\delta_{t+1}$ in (26) is a linear-fractional problem with respect to $\hat{\delta} \in S_{t+1} \subset \mathbb{R}^3$. This problem is reducible to linear programming via a standard change of variables [19] and can be solved online.*

## 6. Simulations and Comments

In this section, the problem of optimal error quantification is illustrated by simulations and some comments.

Consider the unstable and minimum phase plant

$$y_t - 1.9672y_{t-1} + 1.6393y_{t-2} = 2u_{t-1} - 4/3yt - 2 + v_t \tag{31}$$

with the poles $0.6 \pm 0.5i$ and the zero 1.5 (note, that computational complexity of optimal error quantification (25) and (26) is independent of the orders n and m). The total disturbance is modeled in the form

$$v_t = c^w + w_t + 0.1\delta_t^1 |y_{t-\mu}^{t-1}| + 0.1\delta_t^2 |u_{t-\mu}^{t-1}| \qquad (32)$$

with the parameters

$$c^w = 2, \ \mu = 20, \ \delta = (\delta^w, \delta^y, \delta^u) = (1, 0.1, 0.1) \, .$$

We present simulations with the total disturbance $v_t$ of two kinds.

For *random* disturbance and perturbations, the normalized external disturbance $w_t$ and the coefficients $\delta_t^1, \delta_t^2$ in (32) are independent and uniformly distributed on $[-1,1]$.

For *deterministic* disturbance and perturbations, $w_t$, $\delta_t^1$, and $\delta_t^2$ are of the form

$$w_t = cos(5t), \ \delta_t^1 = sin(70t), \ \delta_t^2 = cos(ln(0.5t)) \, .$$

**Comment 1**. The dead zone parameter in simulations was chosen as $\varepsilon = 10^{-4}$, which is three orders of magnitude less than the gains of perturbations $\delta^y, \delta^u$. This choice guarantees high accuracy of the solution of the problem of optimal error quantification. Despite $\varepsilon$ being so small, the number of updates of the estimates $S_t$ and $\delta_t$ in numerous simulations did not exceed 15 on the time interval [1, 1000] and 25 on the time interval [1, 10,000]. The time of each of the numerous simulations on the notebook with the processor i5-7200U CPU @2.50GHz was in the range of 0.6–0.9 s.

**Comment 2**. In all simulations with the optimal closed loop system (1) and (6), the current estimates $\delta_t$ were of the form

$$\delta_t = (\delta_t^w, 0, 0) \, , \qquad (33)$$

which corresponds with zero perturbations in the plant. The values of $J(\delta_t) = \delta_t^w$ were in the interval [3.8, 5.4] in all simulations, while the optimal value was near $J(\delta) = 9.4153$. Possible reasons for the estimates of the form (33) seem to be as follows. First, it can be seen that the values of the control criterion $J(\delta)$ are more sensitive to increasing the gain $\delta^y$ of the output perturbations compared with the upper bound $\delta^w$ of the external disturbance and even more sensitive to increasing the gain $\delta^u$ of the control perturbations. This means that small values of $\hat{\delta}^y$ and $\hat{\delta}^u$ are more preferable in the optimal estimation (26). Second, the total random and deterministic disturbances $v_t$ described above are not the worst-case ones that maximize $\|y - r\|_{ss}$. The problem of modeling the worst-case total disturbance $v_t$ is challenging and its solution is unknown.

To make the disturbance and perturbations (32) worse for the control criterion $\|y - r\|_{ss}$, their values in the time intervals [251, 255], [551, 560], and [751, 770] were chosen in the form

$$w_{t+1} = \delta_{t+1}^1 = \delta_{t+1}^2 = sign(r_{t+1}) \qquad (34)$$

to maximize each subsequent value of $|y_{t+1}|$ on these intervals. The bursts of the tracking error $y_t - r_t$ on all presented plots are the result of this local maximizing $|y_{t+1}|$. Despite the "locally worst" disturbance and perturbations of the form (34), the estimates $\delta_t$ in the closed loop system (1), (6), remained unchanged (33).

Consider now a more realistic situation where the exact nominal model is unknown, but the controller designer has some estimate of it and the task is to evaluate the quality of the corresponding optima controller on real data. Let the approximated model be of the form

$$\hat{a}(q^{-1})y_t = \hat{b}(q^{-1})u_{t-1} + \hat{v}_t, \quad t = 1, 2, 3, \dots,$$

with the parameters

$$\hat{a} = (\hat{a}_1, \hat{a}_2) = (-2, 1.6), \quad \hat{b} = (\hat{b}_1, \hat{b}_2) = (2.1, -1.3) \, .$$

The optimal controller for this model is

$$\hat{b}(q^{-1})u_t = (\hat{a}(q^{-1}) - 1)y_{t+1} + r_{t+1} - c^w. \tag{35}$$

The closed loop system (1) and (35) without perturbations is stable because the roots $-49.5203$, $-12.9108$, and $1.4311$ of its characteristic polynomial lie outside the unit disk of the complex plane. Then, for the tracking error in the closed loop system, we have

$$\begin{aligned} y_{t+1} - r_{t+1} = &\delta^w w_{t+1} + \delta^y \Delta^1(y)_{t+1} + \delta^u \Delta^2(u)_{t+1} + \\ &(\hat{a} - a)(y_t, y_{t-1})^T + (b - \hat{b})(u_t, u_{t-1})^T, \end{aligned} \tag{36}$$

where $a = (a_1, a_2)$, $b = (b_1, b_2)$, and the last two terms at the right-hand side can be considered as additional output and control perturbations compared with those in (7). Then the gains $\hat{\delta}^y$ and $\hat{\delta}^u$ of the output and control perturbations in the closed loop system (1) and (35) can be estimated, respectively, as

$$\hat{\delta}^y \leq \delta^y + \|\hat{a} - a\|_1 = \delta^y + 0.0721, \quad \hat{\delta}^u \leq \delta^u + \|b - \hat{b}\|_1 = \delta^y + 0.1333....$$

Then the closed loop system (1) and (35) is robustly stable. The plots of the tracking errors for the closed loop system (1) and (35) under random and deterministic perturbations are presented on the left panes of Figures 2 and 3, respectively.

**Comment 3**. It was noted in the Introduction that the set-membership approach was criticized for conservatism of prior upper bounds [2]. The optimal error quantification (25) and (26) needs no prior upper bounds except the condition of robust stabilizability (18). This condition is necessary for strict mathematical proof of Theorem 2.

**Comment 4**. Simulations with the "locally worst" disturbance and perturbations (34) of maximal magnitudes and inexact nominal model with the respective non-optimal controller (35) illustrate **additional nonconservatism** of the set-membership approach. Indeed, the unfalsified upper bounds $J(\delta_t)$ on the tracking error were considerably less than the optimal upper bound $J(\delta)$ despite disturbance and perturbations of maximal magnitudes (but not the worst-case for the control criterion (5)). In other words, this approach with optimal estimation is able to take into account a real randomness of disturbance and perturbations. Moreover, simulations with the gains $\delta^y$ and $\delta^y$ that slightly violate the robust stability condition (9) showed similar results and really worst-case perturbations seem to be necessary to obtain estimates $J(\delta_t)$ closer to the worst-case upper bound $J(\delta)$.

**Comment 5**. Despite the estimates of the form (33) associated with no uncertainty in the model, the presented plots for the case of non-worst-case perturbations indirectly provide information about the presence of uncertainty $\delta^y \Delta^1(y)_t + \delta^u \Delta^2(u)_t$ in the closed-loop system. Indeed, if there were no uncertainties in the plant (1), the tracking error should remain in the intervals $[-J(\delta_t), J(\delta_t)]$ in steady-state. But this is not the case on the presented plots.

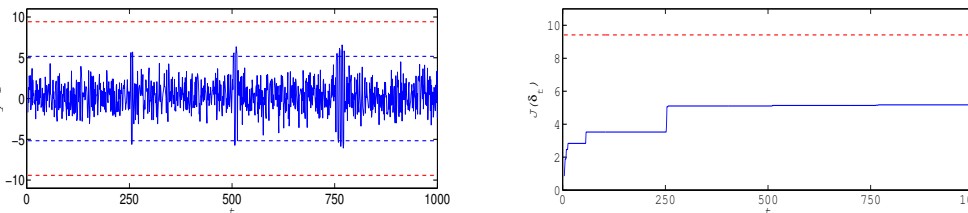

**Figure 2.** Plots of $y_t - r_t$ (**left** pane) and $J(\delta_t)$ (**right** pane) under random $w, \Delta^1, \Delta^2$. $\pm J(\delta)$—red lines, the optimal unfalsified bounds $\pm J(\delta_{1000})$—blue lines.

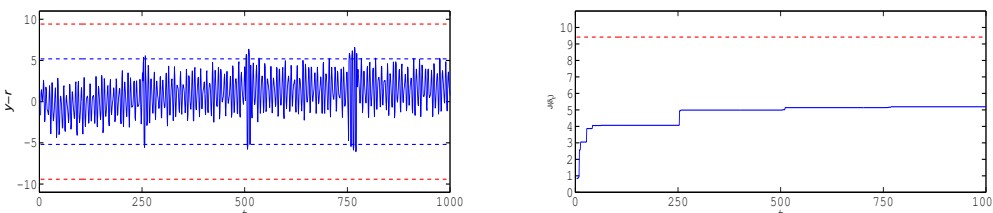

**Figure 3.** Plots of $y_t - r_t$ (**left** pane) and $J(\delta_t)$ (**right** pane) under deterministic $w, \Delta^1, \Delta^2$. $\pm J(\delta)$—red lines, the optimal unfalsified bounds $\pm J(\delta_{1000})$—blue lines.

## 7. Suboptimal Robust Tracking under Unknown Bias of External Disturbance

In this section, we show how the optimal error quantification from section 5 can be used for suboptimal robust tracking under an unknown bias $c^w$ of the biased external disturbance.

Let us introduce more detailed notation,

$$J(c^w, \delta) := J(\delta),$$

for the control criterion $J(\delta)$ defined in (10). Let a priori information on the unknown bias $c^w$ be of the form

$$c^w \in [c^w_{min}, c^w_{max}]$$

with some known $c^w_{min}$ and $c^w_{max}$. Let us choose a natural number N and define the grid of tested values $c^w_k$ of the unknown bias $c^w$ as follows:

$$c^w_k = c^w_{min} + k\varepsilon_1, \ k = 0, 1, \cdots, N, \quad \varepsilon_1 = \frac{c^w_{max} - c^w_{min}}{N}. \tag{37}$$

This grid will approximate the best unfalsified estimate of $c^w$ with the accuracy $\varepsilon/2$. Now, we will use optimal online estimation (24) and (25) for all $c^w_k$ in parallel. The control $u_t$ at the time instant $t$ is computed by the optimal controller (6) corresponding to the bias $c^w_{k_t}$, where

$$k_t = \underset{k}{\operatorname{argmin}} J(c^w_k, \delta^k_t) \tag{38}$$

and $\delta^k_t$ is the estimate of $\delta$ corresponding to $c^w_k$ at the time instant $t$.

Typical results of applying the described adaptive control under parameters

$$c^w = 2.5, \ [c^w_{min}, c^w_{max}] = [-5, 5], \ N = 10, \ \varepsilon_1 = 1$$

and random disturbance and perturbations are presented in Figure 4. Note that $c^w_8 = 2$ and $c^w_9 = 3$ are equidistant from $c^w = 2.5$ and the estimates $c^w_{k_t}$ took only these two values, rarely switching between them and, in particular, $c^w_{k_t} = 2$ on the time interval $[765, 1000]$. The time of each of the numerous simulations with unknown $c^w$ was within the interval of 2–4 s.

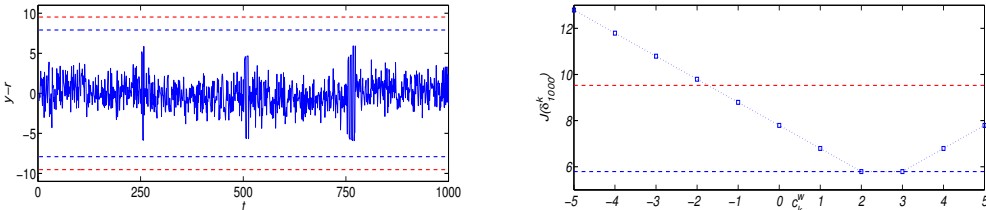

**Figure 4.** Plot of $y - r$ (**left** pane) and $J(\delta^k_{1000})$, $k = 0, 1, \cdots, 10$, (blue squares on **right** pane). $\pm J(\delta)$—red lines, $\pm J(\delta^{k_{1000}}_{1000})$—blue lines.

Note that online estimation of the bias $c^w$ is a nontrivial problem in both classical adaptive control and the tracking problem under consideration. The described parallel error quantification provided the best estimate $c^w_{k_{1000}} = 2$ of the unknown $c^w = 2.5$, but the value of $J(\delta^{k_{1000}}_{1000})$ itself is not the best unfalsified upper bound of $\|y - r\|_{ss}$ corresponding to $c^w_8 = 2$. This is seen on the plot of the tracking error $y - r$ on the left pane of Figure 4. To compute the best one, it is necessary to leave in the set estimate $S_{1000}$ only those inequalities from (24) that correspond to the last estimate $c^w_{k_{1000}} = 2$ and to delete all others.

## 8. Conclusions

The problem of online control-oriented optimal quantification of the unknown bound of biased external disturbance and the gains of coprime factor perturbations of the linear time invariant discrete-time plant with a known nominal model is considered in the framework of the $\ell_1$ robust control theory. Computation of the optimal data-consistent upper bounds under the known bias of external disturbance is reduced to linear programming. This makes it possible to compute optimal estimates online and apply them to optimal robust steady-state tracking under the unknown bias of the external disturbance. The results are illustrated by computer simulations.

**Funding:** This research received no external funding.

**Data Availability Statement:** No new data were created or analyzed in this study. Data sharing is not applicable to this article.

**Conflicts of Interest:** The author declares no conflicts of interest.

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
