# Peer review of "Optimal Error Quantification and Robust Tracking under Unknown Upper Bounds on Uncertainties and Biased External Disturbance"

_mathematics, doi:10.3390/math12020197_

Round 1
Reviewer 1 Report
Comments and Suggestions for Authors
The author has presented a study on the optimal error quantification problem using the L_1 theory, considering bounded external disturbances and signal space. It presents results on robust stability and performance and employs the control criterion for error quantification and adaptive optimal control. The following points must be addressed to make the manuscript publishable:
- Discuss the significance of selecting a small dead-zone parameter in order to ensure finite updates to set estimates. Discuss how this choice affects the overall accuracy and optimality of error quantification.
- Analyze the relationship between the chosen dead-zone parameter and error quantification accuracy, emphasizing how a smaller parameter can improve precision while considering potential trade-offs with computational complexity.
- Compare the effectiveness and applicability of the L_1 theory used in this study with other existing theories for optimal error quantification. Highlight any advantages or limitations associated with using the L_1 theory, especially when considering bounded external disturbances.
- How the robustness of the proposed system is addressed in the presence of bounded external disturbances.
- Analyze the potential trade-offs with computational complexity when considering a smaller parameter for improved precision, exploring ways to mitigate these challenges.
- Compare the effectiveness and applicability of the L_1 theory used in this study with other existing theories for optimal error quantification.
- Specify the importance of real-world application of optimal errors quantification in robust control briefly.
- Clearly define the discrete-time minimum phase plant and its relevance to the study. Specify why considering unknown upper bounds of gains and disturbances.
- Break down the steps in the solution of the optimal error quantification problem more explicitly. Explain the reason behind choosing linear programming and how it makes the problem computationally tractable.
- Include a section that interprets the findings, possibly with practical implications or applications. Discuss the significance of the results in the context of existing literature.
- Simplify complex sentences for better readability without compromising precision. .
English needs a definite refinement. The usage of voice and tenses are inconsistant making the manuscript very difficult to read.
Author Response
Dear Reviewer 1,
Please find enclosed pdf file with my replies to your comments.
Sincerely,
Prof. Victor Sokolov

Reviewer 2 Report
Comments and Suggestions for Authors
The research paper centers on quantifying errors optimally and achieving robust tracking in the presence of uncertainties with unknown upper bounds and biased external disturbances. The simulation results confirm the efficiency of the suggested approach. In general, the study is quite interesting, but the reviewer has some concerns regarding the paper's contributions. These concerns are outlined below.
- Simply introducing the present study in the introduction section is insufficient; it requires thorough expansion and restructuring. Additionally, the authors must explicitly identify the knowledge gaps the research seeks to address, taking into account the existing state of the research problem.
- For the equation (32), please show more details about the derivations.
- The tested plant is small. A large scale system should be used to evaluate the proposed method.
- A comparison with previous research should be added to the results.
-In order to provide a comprehensive assessment of the effectiveness and advantages of the work, it is recommended to offer further explanations and discussions regarding the obtained results.
- The proposed method's drawbacks and potential future work should be described in the Conclusion section.
Author Response
Dear Reviewer 2,
please find enclosed my replies to your comments.
Sincerely,
Prof. Victor Sokolov

Reviewer 3 Report
Comments and Suggestions for Authors
My primary concern is with the empirical results. From my understanding of this work, there is an adaptive estimate of the steady-state error, and so I expected that the estimate would converge to said optimal estimate. However, Figures 2 and 3 instead show the optimal estimate computed over the entire interval (or atleast, that is what it looks like because it is constant). Is this how it is supposed to be implemented and, in that case, is it really online? I suppose I expected there to be a sliding window of sorts or atleast an estimate that changes with time.
Comments on the Quality of English LanguageMinor spelling issues found in the document. Please go through with a spell checker.
Author Response
Dear Reviewer 3,
please find enclosed my detailed replies to your comments.
Sincerely,
Prof. Victor Sokolov

Round 2
Reviewer 1 Report
Comments and Suggestions for Authors
The author has addressed all the comments.
The paper can be accepted after English editing.
Comments on the Quality of English LanguageIt needs improvement.
Reviewer 2 Report
Comments and Suggestions for Authors
I am satisfied with the revision and have no comments.
Reviewer 3 Report
Comments and Suggestions for Authors
The response adequately answered my query.
Comments on the Quality of English LanguageNo major outstanding issues.